# Comment on Samulewski et al. Magnetite Synthesis in the Presence of Cyanide or Thiocyanate under Prebiotic Chemistry Conditions. *Life* 2020, *10*, 34

**DOI:** 10.3390/life11121361

**Published:** 2021-12-08

**Authors:** Pranaba K. Nayak

**Affiliations:** Department of High Energy Physics, Tata Institute of Fundamental Research, Homi Bhabha Road, Colaba, Mumbai 400005, India; pranaba@hotmail.com; Tel.: +91-22-22782367

Samulewski et al. (2020) [1] recently reported the effects of cyanide and thiocyanate ions on the synthesis of magnetite under pre-biotic chemistry conditions using the synthetic seawater 4.0 Gy. They subsequently characterized the synthesized phases by using several competitive and complementary instrumental techniques including FTIR-ATR, XRD, TEM, EDS and ^57^Fe Mössbauer spectroscopy. However, I wish to bring to your kind attention several issues in their analysis of Mössbauer spectroscopic data, important missing parameters and the not-so-correct identification of mineral phases, which has subsequently affected much of the discussion in a significant part of the manuscript, although not affecting the conclusions of the investigation.

The authors synthesized magnetite as well as other associated iron-bearing phases, subsequently analyzed and interpreted the obtained room temperature Mössbauer spectra (presented in Figure S1 [1]), and provided the obtained Mössbauer parameters in Table 3 [1]. It is evident from the XRD patterns (Table 2 [1]), that the MG4P sample contains both magnetite and goethite as the major iron oxide phases, with a ratio of 53.7 to 26.9. The corresponding spectrum in Figure S1 indicates the presence of another iron-bearing phase ferrihydrite as well, along with these two oxide phases. However, the authors have only fitted two sextets and one doublet, leaving out goethite which also showed a sextet pattern at room temperature. The assignment of the second sextet to magnetite as well goethite is erroneous. As magnetite is present in the spectrum that contains both magnetite A (tetrahedral) site as well as magnetite B (octahedral) site, another component corresponding to six-line goethite (a sextet at room temperature) needs to be fitted to the spectrum of MG4P. Hence, a spectrum fit routine should consist of three sextets and a doublet, resulting in a combined fit with a total of 18 lines. In the same way, the spectrum obtained from the MG4SCN sample is required to be fitted with four components and 18 lines, corresponding to two magnetite sextets, a sextet for goethite and a doublet for ferrihydrite. The goethite phase can occur on the surface of the magnetite, but it is a separate and standalone phase and needs to be fitted with an additional sextet in the Mössbauer spectrum [2]. We would like to mention here that the relative abundance of goethite is significantly large (not small) in these two samples as mentioned by the authors on page 10, as is evident from XRD patterns (26.9% in MG4P and 36.2% in MG4SCN). Moreover, the magnetite to goethite ratio goes even further, as the Mössbauer technique only detects the iron-bearing phases, leaving behind other non-iron-bearing phases (XRD detected gypsum and sylvite). The images obtained from transmission electron microscopy (Figure 3 [1]) also corroborate the XRD outcome (Table 2; Figure 2 [1]) with the presence of small-sized goethite crystals with large line width. FTIR-ATR spectra with bands at 795 cm^−1^ and 989 cm^−1^ point to the presence of goethite formation and corroborate the above outcome (Figure 1 [1]). The incorrect fitting of the erroneous number of lines by the authors generated the unacceptable hyperfine parameters (Table 3 [1]) for the MG4P and MG4SCN samples, warranting re-analysis. In addition, the assignment should be clear for each magnetite sextet, conventionally assigned either as tetrahedral (A-site) or octahedral (B-site), but not just as magnetite, as mentioned by the authors since the hyperfine parameters of both the phases are distinct in every respect [3].

There is confusion resulting from incorrect explanations while identifying the goethite phases for the spectrum arising from the MGSCN sample. They have stated, “Mössbauer parameters of the MGSCN sample showed a signal referring to the goethite mineral phase that was presented in the spectra as a sextet with isomeric shift at 0.39 mm s^−1^ and quadrupole splitting at −0.23 mm s^−1^. This result is due to the formation of goethite crystals with small sizes, which generated differentiation in quadrupole splitting and hyperfine magnetic field parameters.” However, the facts are the other way around. The sextet with the above parameters is from well-crystallized goethite with a higher hyperfine field. However, they were silent about the other goethite phase (with isomer shift 0.34 mm s^−1^, quadrupole splitting at −0.02 mm s^−1^ and magnetic hyperfine field of 33.6 T), in which the reduced field is due to the small particle size of goethite, and not in the other goethite fraction (higher hyperfine field of 39.6 T represents the well-crystallized goethite fraction) [2].

For the MGSCN sample, they fitted a large number of Lorentzians (comprising four sextets and one doublet), and they obtained multiple goethite sextets. However, the authors need to provide the line-width of individual sub-spectra, which are essential in the present context as multiple iron oxide phases exist simultaneously. Furthermore, it is essential to report the uncertainty values and the line sharpness parameters, which the authors overlooked. They failed to provide uncertainties in the various hyperfine parameters such as isomer shift, quadrupole splitting and hyperfine field, making it impossible to understand the quality of fitting, validity of their analysis and the basis of their subsequent interpretation [4]. In supplementary materials, the authors mentioned the Mössbauer spectra of 26 samples in (Figure S1 [1]), which we presume to be the six samples as described throughout the paper.

## Data Availability

Not applicable.

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
