# Peer review of "Comment on Samulewski et al. Magnetite Synthesis in the Presence of Cyanide or Thiocyanate under Prebiotic Chemistry Conditions. Life 2020, 10, 34"

_life, 2021, doi:10.3390/life11121361_

Round 1

Reviewer 1 Report

Do the comments affect the major conclusions by the authors of the original article?

Author Response

We thank the learned referee for his time spent carefully reviewing the manuscript. We are also thankful for his in-depth opinion regarding the science and presentation of the material. We have tried to improve certain aspects on the language, readability and clarity front of the comment, as attached file. 

The attached file has been revised with marked up using the “Track Changes” function using MS Word, such that changes can be easily viewed. Please see the attachment.

Reviewer 2 Report

I found interesting to read and check in the original manuscript those critics moved from Pranaba K Nayak to the authors of Samulewski et al.

I think that the best for the authors of Samulewski et al., of both articles is to write an errata corrige in witch the authors can correct that results non well reported previously. This effort will made the reserach of Samulewski et al. strong.

The correct of samples analysed is 6 an not 26 as reported 

I have nothing to add

Author Response

(The authors gave the same response as above.)
